# Cellulose Aerogel Derived Hierarchical Porous Carbon for Enhancing Flavin-Based Interfacial Electron Transfer in Microbial Fuel Cells

**DOI:** 10.3390/polym12030664

**Published:** 2020-03-17

**Authors:** Deng Wang, Ying Wang, Jing Yang, Xiu He, Rui-Jie Wang, Zhi-Song Lu, Yan Qiao

**Affiliations:** 1School of Materials and Energy, Southwest University, Chongqing 400715, China; 2Chongqing Key Laboratory for Advanced Materials & Technologies of Clean Energies, Southwest University, Chongqing 400715, China

**Keywords:** cellulose, porous carbon, hierarchically porous structure, microbial fuel cells, interfacial electron transfer

## Abstract

The flavin-based indirect electron transfer process between electroactive bacteria and solid electrode is crucial for microbial fuel cells (MFCs). Here, a cellulose-NaOH-urea mixture aerogel derived hierarchical porous carbon (CPC) is developed to promote the flavin based interfacial electron transfer. The porous structure of the CPC can be tailored via adjusting the ratio of urea in the cellulose aerogel precursor to obtain CPCs with different type of dominant pores. According to the electrocatalytic performance of different CPC electrodes, the CPCs with higher meso- and macropore area exhibit greatly improved flavin redox reaction. While, the CPC-9 with appropriate porous structure achieves highest power density in *Shewanella putrefaciens* CN32 MFC due to larger active surface for flavin mediated interfacial electron transfer and higher biofilm loading. Considering that the CPC is just obtained from the pyrolysis of the cellulose-NaOH-urea aerogel, this work also provides a facile approach for porous carbon preparation.

## 1. Introduction

The microbial fuel cell (MFC) is a promising environment remediation technology that can degrade pollutants like azo dyes, heavy metal ions in wastewater and generate electricity simultaneously [1]. It also has been used as biosensors for toxicity evaluation of wastewater [2]. It is believed that the micro- or nanostructure dependent interfacial electron transfer between the electroactive microbes and the solid electrode is a key process in the anode that determines the MFC performance [3,4]. As an important endogenous electron mediator, flavin allows bacteria such as *Shewanella spp* to achieve long distance electron transfer to the electrode [5,6,7]. At the same time, the flavins bound to the electrode also play a key role in the direct electron transfer between the outer membrane cytochrome protein and the electrode [8,9,10]. In this case, appropriate porous structure of the MFC anode are essential for the flavin based interfacial electron transfer. In our previous work, it has been demonstrated that the size and curvature of mesopores are critical for the redox reaction of the endogenous flavin mediators in bacteria cellulose derived carbon fibers [11] and sucrose derived ordered mesoporous carbon [12]. However, the effect of hierarchical porous structure with different proportion of meso- and macro-pores on the flavin based interfacial electron transfer process is not very clear.

Biomass-derived activated carbon materials, benefiting from the inherent porous architecture, high conductivity, low-cost and eco-friendly properties, have been testified to be effective electrode in both electrochemical energy storage systems and bioelectrochemical systems [13,14]. However, the porous structure of biomass derived porous carbon often depends on the native structure of biomass precursors and is hard to tailor. Recently, the cellulose aerogel has been widely used to prepare porous carbon, especially the hierarchical porous carbon [15,16,17,18,19] for supercapacitor electrodes or CO_2_ adsorption. Typically, the cellulose aerogel was prepared by dissolving cellulose precursors in NaOH/urea/H_2_O solution at −12 °C, gelling at 40 °C for more than 36 h and subsequent freeze-drying [17,18]. To increase the specific surface area of the obtained porous carbon, a subsequent alkali activation process is also required [15,19], which makes the preparation more complicated and time-consuming. Since the cellulose dissolved in NaOH/urea solution could form stable cellulose/NaOH/urea sol [20], it is possible to skip the gelling step to use the cellulose/NaOH/urea aerogel as carbonization precursor. It has been reported that the urea could accumulate on the cellulose hydrophobic region to prevent dissolved cellulose molecules from re-gathering [21]. In this case, the interspace of the cellulose chain might be tailored by adjust the ratio of urea in the sol.

In this work, cellulose-derived porous carbon (CPC) materials with hierarchical porous structure obtained by dissolving cellulose in an aqueous alkali-urea solution at −12 °C, followed by freeze drying and pyrolysis (Scheme 1). The pore structure and specific surface area of porous carbon can be adjusted by different urea content in the cellulose/NaOH/urea aerogel and a simultaneous alkali activation. The obtained hierarchical porous CPC materials were applied as MFC anode to investigate the pore structure dependent interfacial electron transfer.

## 2. Materials and Methods

### 2.1. Materials

Cellulose (α-cellulose, ≤25 μm, CAS: 9004-34-6), sodium hydroxide (ACS, 97%, CAS: 1310-73-2), urea (ACS, 97 %, CAS: 57-13-6), and lithium hydroxide (anhydrous, 98 %, CAS: 1310-65-2) were obtained from Aladdin Inc., Shanghai, China. Hydrochloric acid (GR) was obtained from CHUANDONG CHEMICAL (Chong Qing, China). The water used in the experiment was deionized water.

### 2.2. Porous Carbon Preparation

Cellulose was used as carbon precursor, which was dissolved in an NaOH/urea/H_2_O solution. In detail, the 4.5 g cellulose was added into pre-cooled NaOH/urea solution to form a 50 g aqueous solution containing the 6 wt. % NaOH and different ratio of urea, which was 6 wt. %, 9 wt. %, and 12 wt. % respectively. After the transparent cellulose sol was obtained, it was freeze-dried and pyrolyzed in a tube furnace at 500 °C for 1 h with a heating rate of 2 °C min^−1^ and 800 °C for 2 h with a heating rate of 5 °C min^−1^ under Ar atmosphere. The obtained samples were washed thoroughly with hydrochloric acid and deionized water in order to remove the residual chemicals. Finally, the porous carbon materials were obtained after drying the prepared samples at 60 °C for 12 h. The obtained carbon materials were named as CPC-n (cellulose-derived porous carbon, CPC; n was the ratio of urea, which was 6 wt. %, 9 wt. %, 12 wt. %, n = 6, 9, 12).

### 2.3. Calculation of Molecular Weight of Cellulose

The molecular weight of the cellulose was calculated with Mark-Houwink equation based on the measured viscosity of cellulose solution [22]. 0.02 g cellulose was dissolved in 4.6 wt % LiOH/15 wt % urea aqueous solution precooled to −12 °C for 30 min. The cellulose solution was then filtered through a 450 *nm* filter to purify. The molecular weight is obtained by viscosity measurement. Intrinsic viscosities of the solutions were obtained by using a capillary viscometer, and these values were converted to viscosity average molecular weight by using the Mark-Houwink equation, [η] = KM^α^, Here, the M_η_ = 2.398 ∗ 10^5^.

### 2.4. Bacteria Culture

A single clone of *Shewanella putrefaciens* (S. *putrefaciens*) CN32 was grown anaerobically in 100 mL of Luria Bertani (LB) broth medium overnight, which was a mixture of 10 g L^−1^ sodium chloride, 10 g L^−1^ tryptone, 5 g L^−1^ yeast extract, and then 10 mL aliquot of bacterial culture suspension was inoculated in 200 mL of fresh LB broth and incubated with shaking at 30 °C until the optical density at 600 nm (OD600) reached about 1.5. The cell pellets were further harvested by centrifuging at 20 °C (6000 rpm, 5 min), and then resuspended in 100 mL M9 buffer (Na_2_HPO_4_, 6 g L^−1^, KH_2_PO_4_, 3 g L^−1^, NaCl, 0.5 g L^−1^, NH_4_Cl, 1 g L^−1^, MgSO_4_, 1 mM, CaCl_2_, 0.1 mM), which was supplemented with 18 mM lactate. The resulted cell suspension was transferred into the anodic chamber of the MFC and purged with nitrogen gas for 30 min to remove the dissolved oxygen before every test.

### 2.5. MFC Set-Up and Operation

Traditional H-type dual-chamber MFC reactors were used in this paper. They consist of an anode chamber, a cathode chamber and a proton exchange membrane (Nafion 117). The carbon cloth with 1 mg cm^−2^ CPC was used as anode and the carbon fiber brush was used as cathode. The cathodic electrolyte was 50 mM potassium ferricyanide in 0.01 M phosphate buffered saline (PBS) (pH 7.4). The polarization and power curves were obtained at the steady state of the MFCs by measuring the stable voltage generated at various external resistances (1–80 kΩ).

### 2.6. Physical Characterization

The morphologies were observed with a field emission scanning electron microscope (FESEM, JEOL 7800F, Tokyo, Japan) and a transmission electron microscopy (TEM, JEOL JEM-2100F, Tokyo, Japan). Before morphology observation, the biofilm adhered anodes were immersed in 4% polyoxymethylene for 12 h and then sequentially dehydrated with ethanol (30%, 40%, 50%, 60%, 70%, 80%, 90%, 100%) and dried in vacuum at room temperature overnight. The Brunauer-Emmett-Teller (BET) surfaces areas and porosity of the samples were evaluated by nitrogen sorption isotherms that measured using an automatic adsorption instrument ASAP 2020 (Micromeritics Instrument, Norcross, Georgia, United States). The thermal stability of CPC precursors was studied using Q50 TGA (TA Instruments NewCastle, DE, USA). The samples were run at the rate of 10 °C/min under nitrogen atmosphere in the range of 30–800 °C. X-ray photoelectron spectroscopy (XPS) measurements were performed on a Thermo Scientific ESCALAB 250Xi electron spectrometer (Waltham, MA, USA).

### 2.7. Electrochemical Characterization

All the electrochemical measurements were conducted with an electrochemical working station (CHI 760E, CHI Instrument, Shanghai, China). The composite-modified CC electrode was used as the working electrode, and the titanium plate and saturated calomel electrode (SCE) were used as counter and reference electrodes, respectively. The electrolyte was 0.1 M phosphate buffer with 2 μM flavin mononucleotide (FMN) or an anaerobic M9 buffer supplemented with *S. putrefaciens* CN32 cell suspension (*S. putrefaciens* CN32 MFC half-cell). The cyclic voltammogram (CV) was recorded at a scan rate of 1 mV s^−1^ after poised at 0.2 V (vs. SCE) for over 72 h. Differential pulse voltammetry (DPV) was performed with a potential step of 4 mV, an amplitude of 25 mV, and a frequency of 1 Hz. Electrochemical impedance spectroscopy was carried out in a frequency range of 0.01 Hz to 100 kHz with a perturbation signal of 10 mV at −0.45 V.

## 3. Results & Discussion

The morphologies of prepared porous carbon materials were examined by FESEM and TEM. From Figure 1, all of the CPC materials exhibited a macroporous foam structure with open and interconnected pores while the carbonized raw cellulose did not show a porous structure (Appendix A). The uniform impregnation of NaOH and urea into cellulose chain apparently promoted the form of porous structure. It is noted that the pore size of CPC-6 seems smaller than that of CPC-9 and CPC-12. The high magnification SEM images (insets) and TEM images show that the pore size of CPC-6 was less than 100 nm while the pore size of CPC 12 was around several hundred nanometers. It indicates that the pore size of the CPCs increased as the amount of the urea in the precursors increased. The enlarged pores could be due to the large amount of gas generated from decomposed urea [23].

The specific surface area and detailed porous structure properties of the CPCs were further examined by using nitrogen adsorption-desorption isotherm analysis. The BET surface area values of three CPCs were compared with that in other reports. As shown in Table 1, the CPCs possessed higher specific surface area than that of the porous carbon without alkali activation process and comparable with that of the activated ones. It is also noted that the CPC-9 possessed highest specific surface area over the three CPCs and the value was higher than that of bamboo cellulose and straw cellulose derived activated carbon in previous reports. From Figure 2a, the three CPCs exhibited type IV isotherms with steep initial region and hysteresis loop, which suggests the existence of micropores, mesopores and macro-pores [24]. This result can be demonstrated by the pore size distribution data calculated from the DFT (Density functional theory) model, which are shown in Figure 2b,c. The volume of micropores was higher than 0.5 m^3^g^−1^ for all CPCs, which contributed a lot to the specific surface area. It is noted that with the increase of urea content, the volume of mesopores, especially the macropores, was greatly decreased. It is possible that the decomposing of excessive urea caused the aggregate of the gas to form large bubbles, which is in agreement with the TEM and SEM results that the dominant pore size of CPCs increased from several tens nanometers to several hundred nanometers as the urea ratio increased. Considering that the micropores were not accessible for flavin mediators [11], the cumulative pore area distribution was also calculated with the BJH (Barrett-Joyner-Halenda) model to evaluate the accessible surface area for three CPC materials. The results (Figure 2d) show that the CPC-9 possessed much higher pore area than the other two materials in small mesopores (2–4 nm) while the CPC-6 possessed higher pore area in large mesopores and macropores. In this case the CPC-9 and CPC-6 may provide a similar active surface for flavin redox reaction, which was larger than that of CPC-12.

The thermal stability analysis results (TG and derivative TG curves) of the CPC precursors are shown in Figure 3a,b. The weight lost at 160 °C~300 °C could be due to the decomposition of urea [26]. It is noted that the decomposition of urea in CPC-6 was slower than that in CPC-9 and CPC-12. The reason might be that the decomposition of excessive urea in CPC-9 and CPC-12, which was not bound in cellulose chain, was faster than that of the bound one. The weight lost at around 300 °C~400 °C could be attributed to the pyrolysis of cellulose together with the urea decomposition [23,26]. The pristine cellulose exhibited fast weight loss at 350 °C~450 °C. It suggests that the intercalation of NaOH and urea into cellulose chain might decrease the thermal stability of cellulose. The XPS results (Figure 3c) show that the CPC-9 and CPC-12 possessed a little bit higher oxygen and nitrogen content than that of CPC-6, which reveals that increase of urea in precursor could promote the doping of nitrogen in the porous carbon. As the nitrogen doped carbon electrodes often exhibited superior performance in the MFC anode [27], the CPC-9 may achieve better bioelectrocatalysis performance than that of CPC-6 in MFCs.

To evaluate the electrochemical behavior of flavins on the CPC electrodes, the cyclic voltammograms (CVs), differential pulse voltammograms (DPVs) and electrochemical impedance spectra (EIS) were measured in 0.1 M phosphate buffer with 2 μM FMN. From the CVs (Figure 4a), the CPC-9 electrode possesses largest capacitive current as well as the redox peak current. It is reasonable as CPC-9 possesses highest specific surface area. The CPC-6 possesses comparable redox peak current with CPC-9 that is higher than that of CPC-12, which has larger specific surface area than the former one. It suggests that the redox peak current of flavins depends on the active surface area rather than specific surface area of the porous electrode. It is further demonstrated by the DPV curves shown in Figure 4b, which exclude the effect of background current [28]. At the same time, the charge transfer resistance of CPC-12 is larger than that of CPC-9 and CPC-6. It suggests that the CPC-6 and CPC-9 electrode could provide the higher active surface area and the faster interfacial charge transfer for flavin redox reaction, which is in agreement with the above discussions about Figure 2d.

To further investigate the electrocatalytic and interfacial electron transfer behavior of the CPC anodes, the electrochemical tests were further conducted in *S. putrefaciens* CN32 half-cells. From the steady state CVs shown in Figure 5a, one catalytic wave with starting potentials at around −0.45 V (vs. SCE) can be observed on three curves, which is attributed to the flavin mediated interfacial electron transfer [29]. It is noted that the CPC-9 had highest catalytic current. In previous reports, the starting potentials of catalytic waves that associated with flavin-dependent indirect electron transfer and cytochrome-dependent direct electron transfer was proposed as −230 mV and 200 mV (vs. SHE) respectively [10]. In this case, the flavins mediated indirect electron transfer should be the main interfacial electron transfer pathway on these CPC anodes. From the DPV curves (Figure 5b), the CPC-6 and CPC-9 have similar flavin redox peak that is higher than that of CPC-12, which is in agreement with the results in Figure 4. It is noted that there was a small peak at around -0.3V vs SCE at the DPV curve of CPC-9 anode, which could be attributed to direct electron transfer occurring via flavocytochrome [10,29]. The Nyquist plots (Figure 5c) reveal that the CPC-9 possessed the smallest interfacial charge transfer resistance. According to the electrochemical analysis results, the CPC-9 possessed the best bioelectrocatalytic performance, which could achieve superior performance in MFCs. The SEM images of the three CPC anodes in *S. putrefaciens* CN32 MFC half-cells suggest that the bacteria loading amount on the electrode is limited. The reason might be that the biofilm formation is still on-going during the short-term half-cell investigation. In this case, the flavin mediated interfacial electron transfer could be the dominant pathway for the extracellular electron transfer of CN32 cells on CPC anodes.

Lastly, the bioelectrocatalytic performances of different CPC anodes were examined in a batch feed MFC for two cycles. The polarization and power density curves were determined by using various external loading resistances (1–80 kΩ) and the results are shown in Figure 6. The CPC-9 exhibited highest maximum power density of 446 mW cm^−2^ while CPC-6 anode and CPC-12 anode delivered a maximum power density of 408 mW cm^−2^ and 381 mW cm^−2^ respectively. Considering that the CPC-6 and CPC-9 possessed similar flavin redox peak current, the superior performance of CPC-9 over CPC-6 in power output might be due to its higher capacitive current [30] or perhaps higher bacteria loading.

After 14 days operation of the MFCs, the surface morphologies of anodes were immediately examined by FESEM. Figure 7 shows that the rod-shaped *S. putrefaciens* CN32 cells adhered on the anodes to form layered biofilm. It seems that there was thicker biofilm on CPC-9 anode than that on CPC-6 and CPC-12 anodes. The protein content on CPC-9 was a little bit higher than that on CPC-6 and CPC-12. As the interfacial electron transfer of the anode mostly relies on the self-generated flavins, the thicker biofilm will guarantee higher interfacial flavin concentration and thus promote the interfacial electron transfer. In this case, the superior power generation performance of CPC-9 anode could be due to its high active surface area for flavin redox reaction, high capacitive current and also the thicker biofilm on the electrode surface.

According to the above results, the function of different pores in hierarchical porous electrode for anodic interfacial electron transfer and power generation in MFCs could be summarized as follows. Both the mesopores and macropores contribute active surface for flavin mediated interfacial electron transfer, which might be the dominant interfacial electron transfer pathway in these CPC-anode MFCs. The micropores would increase the specific surface area as well as the capacitive current of the anode, which could enhance the power density of the MFCs. The micropores might also promote the adsorption of flavins on electrode surface to achieve flavocytochrome based direct electron transfer but it needs further evidence.

## 4. Conclusions

In summary, the cellulose/NaOH/urea aerogel derived hierarchical porous CPC can be obtained through one step pyrolysis and the pore structure could be tailored via adjusting the weight ratio of urea. Increased urea cased enlarged meso- and macropores as well as the doping of nitrogen atom. The CPCs with high meso- and macropore area exhibit greatly improved flavin based interfacial electron transfer. While, the large amount of micropores generated from the reaction of NaOH and carbon during pyrolysis resulted in high specific surface are as well as high double layer capacitance, which enhanced the power generation performance in *S. putrefaciens* MFC. Considering that the CPC is just obtained from the pyrolysis of the cellulose-NaOH-urea aerogel, this work also provides a facile approach for hierarchical porous carbon preparation.

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
