# Peer review of "Cellulose Aerogel Derived Hierarchical Porous Carbon for Enhancing Flavin-Based Interfacial Electron Transfer in Microbial Fuel Cells"

_polymers, 2020, doi:10.3390/polym12030664_

Round 1

Reviewer 1 Report

Manuscript Number:  722201

Cellulose aerogel derived hierarchical porous carbon 2 for enhancing flavin-based interfacial electron 3 transfer in microbial fuel cells

Reviewer(s)’ General Comments to Author:

The performed experimental techniques, obtained and discussed data in this paper should be published in the Polymers after minor revision.

What means with "CPC-6, CPC-9, CPC-12." ?

The NaOH/urea/H2O are standard chemicals. No preference, Authors shoul write the source and the purity of used al chemicals. Also, was he water deionised or not ?

Should be given in Manuscript CAS number for ALL used chemicals.

Physical sizes should be written in italic form.

The manuscript could benefit from more careful proofreading to eliminate the errors due to poor word usage. The authors should be careful with the typographical things. Several errors have been found throughout the paper.

The experiments seem to be carried out carefully and thus the data are reliable. The treatment of data is correct and the obtained results are new and interesting. This paper could be suitable for publication in this journal Polymers while some improvements as mentioned in the letter above have been made.

Reviewer 2 Report

The manuscript describes about utilization of CPC with tailored composition for MFC. The experimental results and discussion are well summarized. However. the authors have already proposed various MFC from carbonized cellulose in ref. 11 and 12. The basic setup of MFC in this study is almost same one. Furthermore, it seems that the presented MFC here does not have some novelty on its function as fuel cell "relative to the previous author's studies". If there is some merit on the MFC presented here, the authors must show clearly in the manuscript, e.g., does tailoring of composition of CPC give some merit for designing function of MFC ? In my opinion, the manuscript cannot be published without a description of novelty of presented MFC system.

Round 2

Reviewer 2 Report

The authors revised the manuscript correctly, especially, an addition of description about function of each pores.

It seems that the worth of revised manuscript is clear via this revision. According to this reason, I recommend an acception of the revised manuscript.